# OpenReview forum: "ReasAlign: Reasoning Enhanced Safety Alignment against Prompt Injection Attack"
_ICLR.cc/2026/Conference — ICLR 2026 Conference Withdrawn Submission_

### Official Review · Reviewer_yypD · 2025-10-27

**Soundness:** 2
**Presentation:** 2
**Contribution:** 2
**Rating:** 2
**Confidence:** 4

**Summary:**

This paper proposes ReasAlign, a model-level defense against indirect prompt injection attacks.
The method fine-tunes an LLM to follow a specific, structured reasoning process. This process involves (1) Problem Analysis, (2) Reasoning to identify conflicts and preserve user intent, and (3) Final Answer Generation.
The goal is to analyze user queries and external data, identify and isolate conflicting malicious instructions, and maintain the user’s original task intent, thereby avoiding the “overkill” problem of prior work (secalign++).
The method is augmented by a “test-time scaling mechanism” , which uses a separate, preference-optimized “judge model” to score and select the best reasoning trajectory from multiple samples.
The authors claim this approach achieves a superior security-utility trade-off, significantly outperforming a baseline (SecAlign++) on benchmarks like CyberSecEval2.

**Strengths:**

- The paper targets a critical and timely vulnerability in modern LLM-based agents: indirect prompt injection.
- The authors correctly identify a significant, practical weakness in existing defenses (which they categorize as “internal defenses” like SecAlign++). This is the “over-defensive” or “overkill” issue, where rigidly suppressing all external instructions severely harms utility when those instructions are benign and necessary.
- The empirical results presented on the CyberSecEval2 benchmark look good, demonstrating a massive utility improvement under attacks (94.6% vs. 56.4%) over the SecAlign++ baseline , while simultaneously achieving a very low Attack Success Rate (3.6%).

**Weaknesses:**

- Baselines: The paper’s “state-of-the-art” (SOTA) claims hinge on outperforming SecAlign++. However, SecAlign++ is not a rigorously peer-reviewed, published method, which may weaken the claim of advancing the SOTA. Besides, the paper dismisses “external defenses” as simple detectors that just “halt task execution”. It fails to compare against a crucial, training-free baseline: a strong, undefended LLM guided by a simple prompt that instructs it to perform the exact same reasoning as ReasAlign. This “strawman” argument means the paper fails to prove its complex method is better than a simple prompt.
- Novelty: The paper’s primary contribution is fixing the “overkill” problem. This appears to be an incremental patch for a fundamental design flaw in prior work (SecAlign++), rather than a novel defensive paradigm.
- Questionable scalability & methodology: The method fine-tunes a fixed reasoning template onto models. This “hard-coded” reasoning path may conflict with or “straightjacket” the more advanced, inherent reasoning capabilities of current flagship LLMs like GPT-5 and Gemini-2.5-pro. There's little evidence the proposed methods are not conflict with model's internal reasonings.
- Unjustified complexity and missing details: The “test-time scaling search”  is a core component of this paper, however, it is vaguely described. The paper mentions a “beam search tree”  but provides no algorithm or figure, making it hard to understand and assess. The overhead analysis in Sec 5.7 is misleading. It only counts final tokens , completely ignoring the significant inference cost of sampling and repeatedly calling the judge model at each step, which doesn't sound like a practical approach.
- Judge model generalization is also unproven: The paper’s claim that its mechanism helps “ensure” logical correctness(Line 271) is a clear overstatement. This judge is trained on a synthetic preference dataset (chosen vs. rejected thoughts). There is no evidence this judge can generalize to correctly score reasoning paths for unseen, complex, or out-of-distribution tasks. The assumption that a judge model (likely to be small) can effectively cover all possible reasoning trajectories for novel tasks is a major, unsupported leap.
- RQs are not presented with clear answers, which is a presentation issue.

**Questions:**

- Missing Baseline Comparison: Why does the paper omit the most critical zero-shot baseline: using a simple prompt to instruct a powerful, general-purpose model (e.g., GPT-5) to perform the exact same reasoning as ReasAlign? How do you justify that your complex training and search method is superior to this simple, training-free alternative?
- Overhead Analysis: The overhead analysis in Section 5.7 only appears to count final output tokens. Does this analysis account for the full end-to-end latency and computational cost of the “test-time scaling search,” including the N sampling steps plus the multiple calls to the “judge model” ?
- Method Scalability: What evidence do you have that your “SFT-instilled,” fixed reasoning template  will not “straightjacket” or conflict with the more advanced, inherent reasoning capabilities of much stronger, frontier models (e.g., at the GPT-5 scale)?
- Judge Model Generalization: Your “judge model” was trained on synthetic preference data. What evidence proves it can generalize to correctly assess the logical correctness of reasoning paths for complex, unseen, or out-of-distribution (OOD) tasks?
- Algorithmic Details: The paper mentions a “beam search tree”  but provides no algorithmic details, pseudocode, or figures, making the method irreproducible. Can you please detail precisely how this search process operates in conjunction with the “judge model” ?

---

### Official Review · Reviewer_b3k7 · 2025-10-27

**Soundness:** 3
**Presentation:** 3
**Contribution:** 3
**Rating:** 4
**Confidence:** 5

**Summary:**

The paper introduces ReasAlign, a model-level defense mechanism designed to mitigate indirect prompt injection attacks in (agentic) large language model systems. Unlike prior approaches that rely mainly on filtering or post-hoc detection, ReasAlign integrates structured reasoning steps directly into the model’s inference process to identify and reject malicious instructions while maintaining the user’s intended task flow.

The authors conduct extensive evaluations across multiple benchmarks, and the results seem good.

**Strengths:**

1. The results are strong - it gets good results on many benchmarks;  compare that to the undefended model or even the previous best defense SecAlign++.
2. The authors did a thorough job testing their approach across many different benchmarks - general tasks, security-specific tests, and agent workflows.
3. This paper tackles a really important security issue that matters more and more as we deploy AI agents in the real world. Indirect prompt injection attacks are a serious threat to these systems, so having a solid defense is crucial for making them safe to use.
4. The idea of using reasoning to detect attacks is reasonable and feels more principled than just adding guardrails. The test-time scaling with a judge model to pick the best reasoning path is an interesting touch that helps ensure the defense actually works reliably.

**Weaknesses:**

1. The paper has some organizational problems in writing. Section 3.2 on threat model seems quite standard and repetitive compared to prior work - it doesn't need to be in the main text. Figure 2 showing prompt injection examples is also unnecessary. More critically, Section 4 on methodology lacks a clear diagram to quickly illustrate how the approach actually works, which would be much more helpful than the redundant threat model description.

2. My main concern is whether this method can scale to larger models. SecAlign++ mainly reports 70B results in their paper, but the authors here seem to only compare against the 8B version, which might be unfair. Looking at Table 1, the 7B model's utility is really bad - so it's hard to tell if ReasAlign's better performance comes from the actual method or just from using GPT-4o-mini as the reasoner while other baselines don't leverage a stronger external model (even it's only for generating structured reasoning steps) . This makes the comparison somewhat unfair and raises questions about scalability.

3. The authors claim reasoning is crucial for their defense, but they only test with the relatively weak GPT-4o-mini. If you used stronger reasoning models like gpt o4 series or Claude-4, wouldn't the results improve significantly? Without any experiments on this, it's unclear how much the reasoning quality actually matters versus just having any reasoning component.

4. Minor issues: The camel citation appears to be wrong; Section 5.6 on node scaling lacks sufficient context - readers unfamiliar with this concept will find it hard to follow.

**Questions:**

Would this method also scale to larger models?

What would happen if stronger reasoning models were used as the reasoner?

---

### Official Review · Reviewer_5q6g · 2025-10-29

**Soundness:** 3
**Presentation:** 3
**Contribution:** 3
**Rating:** 6
**Confidence:** 5

**Summary:**

ReasAlign introduces a reasoning-based safety alignment method that protects large language models from prompt injection attacks without sacrificing utility. It fine-tunes models with LoRA using structured reasoning data—each example guiding the model to analyze inputs, detect malicious instructions, and complete the user’s true task. At inference, multiple reasoning paths are generated and scored by a logic judge trained via Direct Preference Optimization, selecting the most coherent and safe output. Experiments show ReasAlign greatly lowers attack success rates while even improving task performance, demonstrating that reasoning-driven alignment can achieve both strong security and high usability.

**Strengths:**

1. Methodologically simple yet highly effective—built on standard SFT and LoRA without architectural changes.
2. Structured reasoning format (Problem Analysis → Reasoning → Final Answer) provides clear interpretability and controllability.
3. Effectively balances safety and utility, outperforming prior defensive methods.
4. Maintains or even improves general task performance after alignment.

**Weaknesses:**

- Test-time scaling increases inference cost and latency.
- Results rely on LLM-as-judge, which can introduce evaluation bias and reduce accuracy.

**Questions:**

1. Why did the authors rely on an LLM-as-judge to compute ASR instead of using exact tool-call or parameter matching for more objective, reproducible evaluation?
2. Why was the method not evaluated on the ASB [1]?
3. Why did the experiments exclude Qwen models, which are now among the most widely adopted open-source LLMs?
4. In Test-Time Scaling, why does the model generate three reasoning rollouts but the total token usage is not approximately triple? For closed-source models, this would significantly increase inference cost. Would a fairer comparison require baseline models to also perform multiple rollouts and select the best answer for consistency in computational budget?

[1] Zhang et al., *Agent Security Bench (ASB): Formalizing and Benchmarking Attacks and Defenses in LLM-based Agents*, ICLR 2025.

---

### Official Review · Reviewer_o4LU · 2025-11-01

**Soundness:** 2
**Presentation:** 3
**Contribution:** 2
**Rating:** 2
**Confidence:** 4

**Summary:**

This paper proposes a defense mechanism against prompt injection attacks. It first fine-tunes a model (Llama3.1-8B-Instruct) to generate structured reasoning in 3 stages (problem analysis, reasoning, final answer) by distilling a stronger model (GPT-4o-mini). They also propose test-time scaling of best-of-n with a DPO trained judge model. Experiments on benchmarks like CyberSecEval2 show ReasAlign achieves superior performance than undefended LLaMA and SecAlign++

**Strengths:**

This paper tries to address a genuine and important security problem: prompt injection attacks in LLM agents that interact with untrusted external data

Paper is clear in motivation and problem setup

**Weaknesses:**

**Limited technical novelty**

This paper takes very standard approaches like using a stronger model to generate reasoning data, and does best-of-N sampling with a judge model.

**Missing related work and experimental comparisons**

This paper proposes a mix of model-level and test-time scaling approach for prompt injection defense. It only cites 4-5 defense papers (primarily SecAlign series and system-level defenses) while missing significant model-level and detection-based work. In addition, no experimental comparison with these cited work has been provided.
- LLM-LAT Robust [Sheshadri et al., 2024] Latent adversarial training improves robustness to persistent harmful behaviors in llms
- Circuit Breaker [Zou et al., 2024] Improving alignment and robustness with circuit breakers
- SafeChain [Jiang et al., 2025] SafeChain: Safety of Language Models with Long Chain-of-Thought Reasoning Capabilities


**Incomplete analysis**
- Only explores N=1,2,3 (why not 5, 10, 20?). When do diminishing returns occur?
- No analysis of what makes candidates diverse or whether judge actually selects better reasoning
- No comparison to simpler selection strategies (e.g., majority voting, confidence-based selection)
- No investigation of failure modes: When and why does reasoning fail?
- With N=3 candidates + judge scoring, actual cost is >3x but this is never quantified

**Questions:**

The abstract claims you "introduce a test-time scaling mechanism" - but this is standard best-of-N sampling with reward models. Can you clarify what is technically novel beyond applying existing techniques to a new domain?

Why were significant prior works on prompt injection defense omitted in comparison?

---

### Note · Authors · 2025-12-04

I have read and agree with the venue's withdrawal policy on behalf of myself and my co-authors.